# Depression and anxiety among quarantined population during the COVID-19 outbreak in central Ethiopia

Mebratu Abraha[1]*, Getinet Ayano[2,3,4], Dereje Bayissa[5], Abraham Getachew[6], Mahteme Bekele[6], Melsew Getnet[6], Melaku Seyum[7†], Atkure Defar[7], Sileshi Demelash[7], Gizachew Taddesse[7], Tariku Shimels[6]

1 Research Directorate office and Nursing Education Department, Saint Paulo's millennium medical college, Addis Ababa, Ethiopia, 2 Research and Training Department, Amanuel Mental Specialized Hospital, Addis Ababa, Ethiopia, 3 School of Public Health, Curtin University, Perth, Australia, 4 School of Indigenous Studies, the University of Western Australia, Perth, Australia, 5 Nursing Education Department, Saint Paulo's millennium medical college, Addis Ababa, Ethiopia, 6 Research Directorate office, Saint Paulo's millennium medical college, Addis Ababa, Ethiopia, 7 Ethiopian public health institute, Addis Ababa, Ethiopia

† Deceased.

* mebratuabraha21@gmail.com

**Data Availability Statement:** All data are in the manuscript and Supporting information files.

## Abstract

Evidence suggests that quarantine might have a wide-ranging, substantial, and long-lasting negative psychological impact especially when the necessary preventive measures are not taken. This study assessed the prevalence and associated factors of depression and anxiety among quarantined population during the COVID-19 outbreak in central Ethiopia. A community-based cross-section study was conducted among individuals under quarantine from June 5 to July 5, 2020, in Addis Ababa, Ethiopia. The hospital Anxiety and Depression Scale (HADS) was used to assess depression and anxiety. Binary logistic regression analysis (multivariate analysis) was used to identify the potential determinants of depression and anxiety. A total of 297 participants were included in the study which makes the response rate 90.8%. The prevalence of anxiety, depression, and co-morbid anxiety and depression were 21.5%, 70.7%, and 15.8% respectively. In our multivariable analyses, stressful life events (AOR 2.61, 95%CI (1.46, 4.67)), spent time on sleeping (AOR 1.97, 95% CI (1.08, 3.62)), and believing that COVID-19 could be prevented by wearing a glove (AOR 0.30, 95% CI (0.11, 0.81)) showed a statistically significant association with anxiety, whereas being married (AOR 2.67, 95% CI (1.37, 5.22)), had stressful life event in the last six months (AOR 1.44, 95% CI (1.44, 5.25)) and spending of time by sleeping during the quarantine (AOR 1.97, 95% CI (1.42, 6.19)) predicted depression. In conclusion, the current study result indicated that a considerable proportion of individuals who were under quarantine during the COVID-19 pandemic have experienced psychological disturbances, such as anxiety and depression. The results suggest that attention needs to be given to mitigate mental health problems in the quarantined population during the COVID-19 outbreak.

**Funding:** There is no source of funding for the current manuscript.

**Competing interests:** All authors read and approved the final manuscript. The authors declare that they have no competing interests.

**Abbreviations: AOR**, Adjusted Odd Ratio; **CI**, Confidence Interval; **COR**, Crude Odd Ratio; **COVID-19**, Coronavirus disease; **FMOH**, Federal Ministry of Health; **SARS**, Severe acute respiratory syndrome; **SPHMMC**, Saint Paul's Hospital Millennium Medical College; **SPSS**, Statistical Package for Social Sciences; **WHO**, World Health Organization.

## Introduction

Corona Virus Disease 2019 (COVID-19) is an infectious disease caused by a novel coronavirus (2019-nCoV) which was identified as a cause of a group of pneumonia cases in Wuhan, Hubei Province, China in December 2019 [1]. It is highly infectious during the incubation period, and asymptomatic infection might exist. It has been affecting more than 92, 767,845 people resulting in more than 1 986,696 mortalities worldwide [2] and it declared a Global Public Health Emergency (1). Also, in Africa, the virus has spread to dozens of countries within weeks, although governments and health authorities across the continent are striving to limit widespread infections more than one million COVID-19 cases reported [3]. After the first case has been reported on March 13th of 2020 on Japanese citizens in Ethiopia [4], a total of 123,145 detected cases and 1912 deaths has been reported [5]. Due to the rapid and continuous spread of the infection, strong contagion, lethality in severe cases, and no specific medicine, it poses a huge threat to human life and health, and also has a huge impact on the mental health of the general public, causing people to a different degree of psychological problems [6,7]. In Ethiopia, the second populous nation in Africa (110 million people) as of January 14th, 2021, a total of 129,455 confirmed cases and 2,006 deaths have been reported [2].

Since the declaration of coronavirus disease 2019 (COVID-19) outbreak as a pandemic, different countries across the globe have been using quarantine to decrease the transmission of the disease and subsequently to minimize morbidity and associated mortality [8–11]. Evidence suggests that quarantine might have a wide-ranging, substantial, and long-lasting negative psychological impact especially when the necessary preventive measures are not taken [12,13]. The prevalence of psychiatric problems, including anxiety and depression, is high both during and after any infectious outbreaks [14,15]. During the initial phase of the COVID-19 outbreak, one-third of the individuals have moderate-to-severe anxiety and more than half of them rated the psychological impact as moderate-to-severe [16]. These mental health problems could have a long-lasting effect on the mental health of the individual as well as the society, which hinders the urgent response to the current COVID-19 pandemic [14]. Also, individuals with severe depressive symptoms are more susceptible to the severity of the pandemic [17]. In addition, determinants such as female gender, student status, having specific physical symptoms, and poor self-rated health status are significantly associated with a greater psychological impact of the outbreak as well as higher levels of anxiety and depression [16]. Further, factors such as longer quarantine duration, infection fears, frustration, boredom, inadequate supplies, inadequate information, financial loss, and stigma are also other sources of stressors [12]. Whereas, factors like having specific up-to-date and accurate health information about the outbreak situation and particular precautionary measures like wearing personal protective equipment's (PPE) in the reverse has a significance association with a lower psychological impact of the outbreak and lower levels of anxiety, and depression [16].

Factors like a lower level of psychological impact and better mental health status can be used to formulate psychological interventions to improve the mental health of vulnerable groups during the COVID-19 epidemic [16]. Taking appropriate psychological measurements on an individual under quarantine could reduce the negative mental impact [12]. Because the psychological impact of stressful events related to an infectious disease outbreak may be mediated by enhancing peoples' perceptions of those events [18]. Therefore, raising public awareness of the disease and providing positive psychological programs in the media aimed at controlling stress can reduce anxiety in society [7]. Governments and healthcare authorities should take urgent actions to protect the mental health of their community [14] and should focus more on providing economic and medical support to improve the general population's mental state [15].

However, this is not to suggest that quarantine should not be used because the psychological effects of not using quarantine and allowing the disease to spread might be worse [19]. Whereas, whenever quarantine is needed, depriving people of their liberty for the wider public good is often contentious and needs to be handled carefully and officials should take every measure to ensure that this experience is as tolerable as possible for people [12]. Further, the provision of mental health first aid training and services during the quarantine period has paramount significance and benefit to individuals expected to be under quarantine [13]. So, responsible health officials charged with implementing quarantine, are in employment and usually with reasonable job security, should also remember that not everyone is in the same situation [12].

In general, even if the government of Ethiopia in collaboration with other stakeholders has been trying to control and handle the COVID-19 outbreak, there is a gap in information that to what extent does those individuals experiencing psychological problems. Even though the response to quarantine should include assistant for those in quarantine ─ this includes screening to identify risky people, coordinated and consistent provision of COVD-19 related knowledge and attitude as well as basic psychological support to those in quarantine. However, this basic support for quarantine population is not undertaken in Ethiopia, which may increase the risk of mental health people in those population groups. So, the current study proposed with the aim to fill the gap by providing information about the magnitude of depression and anxiety among individuals under quarantine. Then the finding of the current study may also serve as a scientific reference to the health institutions and Ethiopian FMOH to offer comprehensive psychological support to individuals as well as their general population to minimize the psychological impact of every epidemic event and the mental problems. It may also serve as a reference for future studies and to develop appropriate policies and plans by the responsible health and other sectors. These include identification support for those vulnerable groups and psychological support to reduce the stress associated with adverse mental health outcomes.

## Methods and materials

### Study area, period, and design

A community-based cross-sectional study was conducted from June 05 to July 05, 2020, at selected quarantine centers in Addis Ababa, Ethiopia. Addis Ababa is the capital city of Ethiopia, occupying a total area of 540 km$^2$. The city is divided into ten sub-cities and 116 suburbs (Woredas). Initially, during the onset of COVID-19, the Ethiopian government shut down all Universities. And, all campuses of Addis Ababa University, Addis Ababa Science and Technology University, governmental health colleges, and some selected hotels were assigned as quarantine centers. The number of individuals under quarantine varied from time to time, because it depends on the number of graduated individuals from the quarantine centers, travels to other regions, and a number of travels to Ethiopia from abroad. There were around 800 individuals under quarantine in Addis Ababa during the data collection period. During the study (July 2021), Ethiopia has implemented 14 days of mandatory hotel quarantine for all travelers who are permitted to cross the border.

### Inclusion and exclusion criteria

Those participants who were quarantined at Addis Ababa (central Ethiopia) during the study period were included in this study. Those who had health or other problem that inhibit effective communications were excluded from the study.

## Study population

The study population included all individuals under quarantine who were included in the sample.

## Sample size determination and sampling technique

The sample size was determined by using a single population proportion formula considering and assuming the proportion of depression and anxiety among individuals under quarantine to be 50%, a 95% confidence level, and a 5% degree of precision, the required sample size was estimated to be 384.

After adding a non-response rate of 10%; the final sample size was 422. Around 1000 individual were expected to be under quarantine every month, so by considering this one we used the correction formula that is implemented when the source/study population are less than 10000, and the final sample size becomes 297 (adding 10% = 327). This was the representative sample size for the current survey (n = 327).

Regarding the sampling technique, first, we purposively selected and included all quarantine centers found in Addis Ababa, Ethiopia (quarantine centers outside the capital city, Addis Ababa were not included). A simple random sampling technique was employed in order to select a representative sample of participants from the centers. The room numbers and the number of people at each center were taken from the responsible authorities handling this condition and control the overall situation of the quarantine centers ─ such as [Ethiopian Public Health Institute (EPHI) & the Federal Ministry of Health (FMOH)]. Then, we used a lottery method to select individuals from each center.

## Data collection tools, procedure, management, and analysis

A structured questionnaire containing data on socio-demographic characteristics (educational status, age, sex, income, residence, religion, and marital status), quarantine and clinical factors (such as chronic medical conditions and source of information about the quarantine center), psycho-social and substance-related factors [current (one month), six month, and life time use of alcohol, tobacco, cannabis and amphetamines] was used to collect information from the participants. These were relevant factors that have been relatedly linked with anxiety and depression in quarantine people in previous studies [12,16,20].

Depression and anxiety were assessed by using the Hospital Anxiety and Depression Scale (HADS) [21]. HADS is a 14 item questionnaire, extensively used to use to screen for depression and anxiety in Ethiopia [22–24]. The 14 items can be separated into two 7 item subscales for anxiety and depression. It is validated in Ethiopia and internal consistency is 0.78 for anxiety, 0.76 for depression, and 0.87 for full scale. The scale uses a cut-off score for depression and anxiety of greater than or equal to 8 [25].

Social support was measured by Oslo on the 3 items social support scale (OSS-3), which has been commonly used to assess social support and had been used in several studies [26,27]. It has a score ranging from 3–14, which had three broad categories: "Poor Support" 3–8, "Moderate Support" 9–11, and "Strong Support" 12–14.

Self-reported and structured questionnaires were used to assess other factors ─such as quarantine related and clinical factors, sociodemographic, psycho-social and substance-related factors.

Data collectors who have had prior experience and background in health sciences were involved (BSc in psychiatry/BSC in nursing). They were selected and trained by the principal investigator. Eight data collectors and three supervisors were involved in the data collection and supervision process, respectively. The principal investigator supervised the whole activities

of the data collection process. The training was delivered to data collectors and supervisors on how to use the questionnaire, the ethical principles of confidentiality, and data management before their involvement with data collection.

### Data analysis

All questionnaires were checked for completeness and consistency of responses manually by the supervisors and researchers. After cleaning, data was entered into EPI-Data version 3.4 and was exported to SPSS version 25 for analysis. Descriptive statistics (frequencies and percentages) were used to explain the study participant in relation to study variables. Bivariate and multivariate analyses were performed to determine the presence of statistically significant associations between the independent and dependent variables. Those variables with a p-value <0.02 in the bivariate analysis were selected to be included in the multivariable logistic regression to control for possible confounders. Additionally, we have used Bradford Hill's causal criteria including the consistency of the association to the select important variables to include in the multivariable model [28]. The strength of the association was presented in the odds ratio and 95% confidence interval. A p-value of $< 0.05$ on multivariate analyses was considered statistically significant.

### Ethical considerations

Ethical approval and clearance were obtained from Saint Paul's Hospital Millennium Medical College (SPHMMC) ethical review board (Reference number: 1m23/81). In addition, oral consent of the respondents was obtained after giving information and thoroughly explaining the aim of the study to each respondent. The participants willingness was asked by a self-reported question (yes or no) and documented together with other questions by the data collectors. Due to COVID-19 related restrictions, we did not use written consent and this has been approved by IRB. The respondents were told that participation is voluntary, could withdraw at any time, or refuse to answer any question if they want to. No information concerning the individual was passed to a third party.

## Results

### Socio-demographic characteristics of the respondents

A total of 297 participants were included in the study. The response rate was 90.8%. The mean age of the respondents was 28.33(SD = 6.22) years. More than one-third of the participants (38.6%) were young adults (ages 18–25 years). Approximately one-fourth of the participants were females (73.1%) and more than half were unmarried (56.9%) and Muslims by religion (51.5%). About 40.1% of the participant had attended primary education and the average income for the participants was 8783.74 Ethiopian birr (251.44 US dollars). (Table 1). Almost all the participants have been living abroad (99.3%).

### Clinical and psychosocial characteristics

The majority of the participant (90.9%) had a chronic illness and more than half spend their time on social media at the quarantine center (53.5%). Roughly half of the participants (49.8%) had intermediate social support and more than half (58.2%) had no stressful life events in the last 6 months. Regarding substance use history, approximately one-fifth of the participants had used tobacco (21.2%) and alcohol (18.2%) in their lifetime (Table 2).

   **Awareness of COVID-19 pandemic and PPE utilization.** The current study also showed that the vast majority of the respondents (282, 94.9%) had heard about the COVID-19

**Table 1. Descriptions of socio-demographic factors among individuals under quarantine during the Covid-19 pandemic, in Addis Ababa, Ethiopia, 2020.**

| Variables | | Frequency | Percent (%) |
|---|---|---|---|
| **Age** | 25 and below | 114 | 38.6 |
| | 26–30 | 96 | 32.5 |
| | 31 and above | 85 | 28.8 |
| **Sex** | MALE | 79 | 26.6 |
| | Female | 217 | 73.1 |
| **Marital status** | Married | 105 | 35.4 |
| | Single | 169 | 56.9 |
| | Divorced | 23 | 7.7 |
| **Religion** | Orthodox | 89 | 30.0 |
| | Muslim | 153 | 51.5 |
| | Protestant | 53 | 17.8 |
| | Catholic | 2 | .7 |
| **Educational status** | Unable to read and write | 22 | 7.4 |
| | Able to read and write | 31 | 10.4 |
| | Primary school | 119 | 40.1 |
| | Secondary school | 105 | 35.4 |
| | Certificate and above | 20 | 6.7 |
| **Job** | Government employed | 1 | .3 |
| | Student | 2 | .7 |
| | Merchant | 19 | 6.4 |
| | Unemployed | 33 | 11.1 |
| | Other specify (maid) | 241 | 81.1 |
| **Residence** | Urban | 139 | 46.8 |
| | Rural | 158 | 53.2 |
| **Have a child** | Yes | 125 | 42.1 |
| | No | 172 | 57.9 |
| **Living with family** | Yes | 6 | 2.0 |
| | No | 291 | 98.0 |
| **Were living abroad** | Yes | 295 | 99.3 |
| | No | 2 | .7 |
| **Income** | 8000 and below | 209 | 70.4 |
| | >8000 | 88 | 29.6 |

pandemic which roughly three fourth reported their source was from mass media (74.1%). Two hundred six (69.4%) and 183 (61.6%) considered that COVID-19 is a disease of the respiratory tract and caused by a virus respectively. In addition, almost all agreed that COVID-19 can be prevented by using a facemask (98.3%), wearing gloves (81.8%), using a sanitizer (88.2%), and keeping a physical distance (98.3%). Concerning COVID-19 disease prevention methods utilization, 286 (96.3%) reported that they were using a facemask, 281 (94.6%) keep their physical distance, 253 (85.2%) wash their hands, 151 (50.8%) use a sanitizer whereas, only a small proportion (37.4%) have used gloves. (Table 3).

**The prevalence of anxiety, depression, and comorbid.** This study revealed that the prevalence of anxiety, depression, and co-morbid anxiety and depression was 21.5% (CI 16.8, 26.6), 70.7% (CI 65.7, 76.1), and 15.8% (95% C.I: 11.8, 20.2) respectively

**Factors associated with anxiety and depression.** *Bivariate Analysis.* In bivariate analysis, socio-demographic factors such as having a child, stressful life events, spending the quarantine

**Table 2. Description of clinical, psychosocial & substance use factors among individuals under Quarantine for the case Covid-19 Pandemic, at Addis Ababa, Ethiopia, 2020.**

| Variables | | Frequency | Percent (%) |
|---|---|---|---|
| **Reason for quarantine** | History of traveling to the COVID-19 outbreak area | 297 | 100.0 |
| **Diagnosed chronic illness** | Yes | 27 | 9.1 |
| | No | 270 | 90.9 |
| **Type of co-morbid** | Diabetes | 2 | .7 |
| | Renal disease | 7 | 2.4 |
| | Hypertension | 11 | 3.7 |
| | other specify | 6 | .7 |
| **Method of refreshment at the quarantine center** | Watching TV | 6 | 2.0 |
| | Listening to a radio | 7 | 2.4 |
| | Using social media like Facebook, YouTube etc.. | 159 | 53.5 |
| | Reading a book or other published documents | 26 | 8.8 |
| | by sleeping | 99 | 33.3 |
| **social support** | poor social support | 114 | 38.4 |
| | intermediate support | 148 | 49.8 |
| | good social support | 35 | 11.8 |
| **Stressful life events** | No | 173 | 58.2 |
| | Yes | 124 | 41.8 |
| **Ever use of tobacco products** | Yes | 63 | 21.2 |
| | No | 234 | 78.8 |
| **Ever use of alcohol** | Yes | 54 | 18.2 |
| | No | 243 | 81.8 |
| **Ever use of Amphetamine** | Yes | 82 | 27.6 |
| | No | 215 | 72.4 |
| **Tobacco within a month** | Yes | 35 | 11.8 |
| | No | 262 | 88.2 |
| **Alcohol use within a month** | Yes | 14 | 4.7 |
| | No | 283 | 95.3 |
| **Amphetamine use within a month** | Yes | 4 | 1.3 |
| | No | 293 | 98.7 |

time by sleeping, and perceiving transmission of COVID-19 can be prevented by wearing of glove for anxiety were found to be significantly associated with anxiety in quarantine people (Table 4). Whereas, marital status, stressful life event, method of refreshment, source of information about COVID-19, perception on the cause of COVID-19, and knowledge on method of prevention and utilization of COVID-19 prevention methods for depression were found to be significantly associated with depression among the participants (Table 5).

*Multivariate Logistic Analysis.* In our multivariate analysis; factors such as those who had stressful life events in the last six months (AOR 2.61, 95%CI (1.46, 4.67)), who spent their time sleeping during the quarantine (AOR 1.97, 95% CI (1.08, 3.62)), and those who considered COVID-19 could not be prevented by wearing gloves AOR 0.30, 95% CI (0.11, 0.81)) were statistically significant association with anxiety (Table 4).

Whereas being married (AOR 2.67, 95% CI (1.37, 5.22)), had stressful life events in the last six months (AOR 1.44, 95% CI (1.44, 5.25)), and spending of time sleeping during the quarantine (AOR 1.97, 95% CI (1.42, 6.19)) were significantly associated with depression (Table 5).

**Table 3. COVID-19 related awareness and PPE utilization behavior of the study participants in at quarantine centers, in Addis Ababa, Ethiopia, 2020.**

| Variables | | Frequency | Percent (%) |
|---|---|---|---|
| **Hear about the COVID-19 pandemic** | Yes | 282 | 94.9 |
| | No | 15 | 5.1 |
| **Source of information** | Mass media | 220 | 74.1 |
| | Health professional | 7 | 2.4 |
| | From my family | 14 | 4.7 |
| | from my friends | 24 | 8.1 |
| | Other | 17 | 5.7 |
| | Hypertension | 11 | 3.7 |
| | other specify | 6 | .7 |
| **COVID-19 is caused by** | Bacteria | 17 | 5.7 |
| | Virus | 183 | 61.6 |
| | Fungus | 5 | 1.7 |
| | Parasite | 3 | 1.0 |
| | I don't know | 88 | 29.6 |
| | Other | 1 | .3 |
| **COVID-19 is** | Disease of the heart | 25 | 8.4 |
| | Disease of the respiratory system | 206 | 69.4 |
| | I don't know | 66 | 22.2 |
| **Which one of the following will help us to prevent COVID-19** | | | |
| handwashing | Yes | 284 | 95.6 |
| | No | 13 | 4.4 |
| Face mask | Yes | 292 | 98.3 |
| | No | 5 | 1.7 |
| By wearing gloves | Yes | 243 | 81.8 |
| | No | 54 | 18.2 |
| By using sanitizer | Yes | 262 | 88.2 |
| | No | 35 | 11.8 |
| By keeping physical distance | Yes | 292 | 98.3 |
| | No | 5 | 1.7 |
| **Did you use the following to prevent COVID-19** | | | |
| Face mask | Yes | 286 | 96.3 |
| | No | 11 | 3.7 |
| Glove | Yes | 111 | 37.4 |
| | No | 186 | 62.6 |
| Sanitizer | Yes | 146 | 49.2 |
| | No | 151 | 50.8 |
| Hand washing | Yes | 253 | 85.2 |
| | No | 44 | 14.8 |
| Keep physical distance | Yes | 281 | 94.6 |
| | No | 16 | 5.4 |
| **COVID-19 affect all age groups** | Yes | 239 | 80.5 |
| | No | 58 | 19.5 |

## Discussion

This work represents one of the few studies that explored the prevalence and factors associated with anxiety and depression among in quarantined population during the COVID-19 outbreak in Sub-Saharan Africa including Ethiopia. Our final analysis revealed a high prevalence

**Table 4. Factors associated with anxiety among individuals under Quarantine for the case Covid-19 Pandemic, at Addis Ababa, Ethiopia, 2020.**

| Explanatory Variables | Anxiety | | COR,95%(CI) | AOR,95%(CI) | p-value |
|---|---|---|---|---|---|
| | No | Yes | | | |
| **Have a child** | | | | | |
| No | 128 | 44 | 1 | 1 | |
| Yes | 105 | 20 | 1.81, (1.00, 3.25)* | 1.53, (0.82, 2.84) | 0.181 |
| **Stressful life event** | | | | | |
| Yes | 85 | 39 | 2.72, (1.20, 8.25)* | 2.61, (1.46, 4.67)** | 0.001 |
| No | 148 | 25 | 1 | 1 | |
| **Methods refreshment** | | | | | |
| By sleeping | 70 | 29 | 4.97, (1.10, 3.40)* | 1.97, (1.08, 3.62)** | 0.028 |
| Other[1]* | 163 | 35 | 1 | 1 | |
| **COVID-19 can be prevented by wearing gloves** | | | | | |
| Yes | 184 | 59 | 1 | 1 | |
| No | 49 | 5 | 3.14, (1.20, 8.25)* | 0.30, (0.11, 0.81)** | 0.018 |

* Significant association (p-value < 0.05 in bivariate).

**-significant association (p-value<0.05 in multivariate analysis) Hosmer and Lemeshow test = 0.027.

[1]*other methods of refreshment = Using social medias (like Facebook, YouTube etc..), Watching TV, listening a radio and Reading a book or other published documents.

of anxiety, depression, as well as comorbid anxiety and depression among the participants. The prevalence of anxiety in this study (21.5%) was higher than the previous study finding conducted in southwestern China (8.3%) [29] and lower than other studies conducted in different countries including China (28.8–58%) [16,17], Huwan (70.78%) [30], Sierra Leone (48%) [31], Australian (34%) [32] and the global prevalence based on finding from a recent meta-analysis (45%) [14]. The discrepancy in the finding of the current study and the previous studies could be the difference in factors influencing the prevalence of anxiety such as sample size and instrument used [14,16,17,29,31,32]and differences in characteristics of the study participant across the studies [14,31,32].

Regarding associated factors of anxiety, in the current study, those who had stressful life events in the last six months were 2.61 times more likely to have anxiety symptoms as compared to those who hadn't. The psychological impact of stressful events could have related to an infectious disease outbreak [18]. Further, factors such as frustration, inadequate supplies, and financial loss are among the different stressful life events which could provoke individuals to experience anxiety [12].

Individuals who spent their time sleeping during the quarantine period were 1.97 times more likely to experience anxiety symptoms based on the hospital anxiety and depression scale as compared to those who spent their time using different social media and mass media. This might be because receiving different refreshing information and games through these social media's as well as mass media's could help the individuals to experience more relaxation and exposure with others. Lack of social support or rejection of oneself from social aspects could expose people to develop more psychological impacts like anxiety [14].

Moreover, those who considered COVID-19 infection couldn't have been prevented by wearing the glove were more likely to have anxiety as compared to those who considered it could prevent. This could be associated with the type of information they received and their way of interpretation. But the current study finding is in contrast with others that inadequate information [31] with a high level of stress, whereas having specific up-to-date and accurate

**Table 5. Factors associated with depression among individuals under quarantine during the COVID-19 pandemic, in Addis Ababa, Ethiopia, 2020.**

| Explanatory Variables | Depression | | COR,95%(CI) | AOR,95%(CI) | p-value |
|---|---|---|---|---|---|
| | No | Yes | | | |
| **Marital status** | | | | | |
| Single | 63 | 106 | 1 | 1 | |
| Married | 18 | 87 | 2.87, (1.58, 5.21)* | 2.67, (1.37, 5.22)** | 0.004 |
| Divorced | 6 | 17 | 1.68, (0.63, 4.49) | 1.44, (0.50, 4.16) | 0.503 |
| **Stressful life event** | | | | | |
| Yes | 23 | 101 | 2.58, (1.49, 4.46)* | 1.44, (1.44, 5.25)** | 0.002 |
| No | 64 | 109 | 1 | 1 | |
| **Methods of refreshment** | | | | | |
| By sleeping | 14 | 85 | 3.55, (1.88, 6.69)* | 1.97, (1.42, 6.19)** | 0.004 |
| Other[1]* | 73 | 125 | 1 | 1 | |
| **Source of information** | | | | | |
| From health professional | 4 | 3 | 1 | 1 | |
| Mass media | 61 | 159 | 3.48, (0.76, 15.98) | 2.46, (0.43, 13.92) | 0.310 |
| From family | 1 | 13 | 17.33,(1.39, 216.60)* | 8.68, (0.60, 126.16) | 0.114 |
| from friend | 12 | 12 | 1.33, (0.24, 7.28) | 0.52, (0.71, 3.84) | 0.524 |
| Other | 3 | 14 | 6.22,(0.89, 43.66) | 1.58, (0.16, 15.58) | 0.693 |
| **Cause of COVID-19** | | | | | |
| Virus | 66 | 117 | 1 | 1 | |
| I don't know | 16 | 72 | 2.54, (1.37, 4.72)* | 1.64, (0.68, 3.95) | 0.274 |
| Other | 5 | 21 | 2.37, (0.85, 6.58) | 2.68, (0.89, 8.06) | 0.080 |
| **COVID-19 can be prevented by wearing gloves** | | | | | |
| Yes | 78 | 165 | 1 | 1 | |
| No | 9 | 45 | 2.36, (1.00, 5.08)* | 1.80, (0.63, 5.13) | 0.271 |
| **COVID-19 can be prevented by using sanitizer** | | | | | |
| Yes | 82 | 180 | 1 | 1 | |
| No | 5 | 30 | 2.73, (1.02,7.30)* | 2.07, (0.43, 9.93) | 0.362 |
| **Did you use sanitizer** | | | | | |
| Yes | 52 | 94 | 1 | 1 | |
| No | 35 | 116 | 1.83, (1.04, 3.05)* | 1.57, (0.83, 2.99) | 0.169 |

* Significant association (p-value < 0.05 in bivariate).

**-significant association (p-value<0.05in multivariate analysis) Hosmer and Lemeshow test = 0.022.

[1]*other methods of refreshment = Using social media (like Facebook, YouTube, etc..), Watching TV, listening to a radio and Reading a book or other published documents.

health information and particular precautionary measures like gloves are associated with a lower psychological impact of the outbreak [32].

Regarding the prevalence and factors associated with depression among individuals under Quarantine for the case Covid-19 Pandemic, the current study result determined that the prevalence of depression was 70.7% which is high. The high prevalence might be related to being separated from close families and friends because social isolation and lockdown can lead to psychological disturbance [33]. Quarantine may have a detrimental impact on mental health which can increase psychiatric symptoms and mental health problems in the general population [34]. Thus, even if quarantine has a wide-ranging, substantial, and long-lasting psychological impact, it is not to suggest that quarantine should not be used; because the psychological effects of not using quarantine and allowing the disease to spread might be worse [35]. But, it

is better to recognize its psychological impact as well as the rapid spread of the COVID-19 pandemic as a public health priority for both authorities and policymakers who should rapidly adopt clear behavioral strategies to reduce the burden of disease and the dramatic mental health consequences of this quarantine related to the current pandemic [36]. Our finding is higher than the reported results from studies which were conducted in China (14.6–27.9%) [16,29,37], Hong Kong (19%) [38], Huwan (26.47%) [30], and the global prevalence on finding from a meta-analysis (28%) [39]. Further, it was also quite higher than study results of different countries like Sierra Leone that any anxiety-depression symptom was reported among 48% of the general population [31], around 15% of the university students living in Bangladesh [40], 6.21% in Shenzhen [41], in Albania 6.22% and 6.28% of bachelor and master university students (nurses/midwives) and their family members' respectively reported symptoms of depression [13] and as per the systematic rapid review and meta-analysis study result, 38% of the healthcare workers (HCWs) were reported depression symptoms [14]. The differences in the methodologies used and the characteristics of the participants might be the possible reasons for the observed variations.

Concerning the associated factors of depression those who were married are 2.67 times more likely to have depression symptoms based on HAD data collection tool that we used as compared to those who were single. This could be due to separation from their sexual partner and more worry about their family. Quarantine experience has a positive correlation with depressive symptoms, while home quarantine is positively correlated with happiness [17].

In addition, those who had experienced stressful life events in the last six months had 1.44 times more likely to have depression as compared to those who hadn't. Stressful life events by itself it has their contribution in the development or exposing individuals to experience more depression symptoms. Being more worried about being infected, having no psychological support, greater property damage, and lower self-perceived health condition has a significant association with higher scores of self-rating depression scale [29]. Also, having specific physical symptoms, principal financial problems or loss [32], infection fears, frustration, inadequate information, and stigma has a great risk of developing psychological problems like depression [12].

With respect to methods of refreshment or way spending time during the quarantine period, the current study found that those who were spending their time by sleeping were 1.97 times more likely to have a high score of depression symptoms as compared to those who were spent their time by using social media (like Facebook, YouTube, etc..), watching TV, listening radio and reading a book or other published documents. But this was in contrast with the previous study report because it was mentioned frequent exposure to social media as a risk factor for depression [42].

### Strength and limitation of the study

The current study had several strengths: First, large sample size and it is one of the few studies on the subject. Second, we have used standard instruments to measure anxiety and depressive symptoms.

The limitations include: due to the nature of the study (cross-sectional study design) the association between the different variables with anxiety and depression may not imply causality. Recall bias is another limitation of the study since we have retrospectively assessed most exposure variables including a history of chronic medical illness and substance use problems.

### Conclusion

In summary, in this study, more than one-fifth and about three-fourth of the participants had anxiety and depression respectively, suggesting anxiety and depression requires attention in

the quarantined population during the COVID-19 outbreak. Moreover, roughly one-sixth of the participant had comorbid anxiety and depression. The presence of stressful life events, time spent on sleeping, and believing that COVID-19 could be prevented by wearing masks were found to be significant predictors of depression or anxiety among the participants. The results suggest that attention needs to be given to mitigate mental health problems in the quarantined population during the COVID-19 outbreak. Screening for mental health issues and providing mental health (psychological) support are needed for the quarantine population.

## Supporting information

**S1 Data. Minimal data set.**
(SAV)

## Acknowledgments

The authors acknowledge saint Paul Hospital Millennium Medical College IRB of the institution for their ethical evaluation and approval. Also, for the institution Academic and Research Vice provost and research directorate office for sponsoring the research proposal. The authors appreciate the respective study institution and the study participants for their cooperation in providing the necessary information.

Dr. Melaku Seyum passed away before the submission of the final version of this manuscript. The corresponding author, Dr. Mebratu Abraha, accepts responsibility for the integrity and validity of the data collected and analyzed.

## Author Contributions

**Conceptualization:** Mebratu Abraha, Getinet Ayano, Mahteme Bekele, Atkure Defar, Sileshi Demelash, Gizachew Taddesse, Tariku Shimels.

**Data curation:** Mebratu Abraha, Melsew Getnet.

**Formal analysis:** Mebratu Abraha.

**Investigation:** Mebratu Abraha, Sileshi Demelash.

**Methodology:** Mebratu Abraha, Getinet Ayano, Dereje Bayissa, Abraham Getachew, Melaku Seyum.

**Project administration:** Mebratu Abraha.

**Resources:** Mebratu Abraha, Melsew Getnet, Gizachew Taddesse, Tariku Shimels.

**Software:** Mebratu Abraha, Melaku Seyum, Sileshi Demelash.

**Supervision:** Mebratu Abraha, Getinet Ayano, Mahteme Bekele, Melsew Getnet, Gizachew Taddesse, Tariku Shimels.

**Validation:** Mebratu Abraha, Abraham Getachew.

**Visualization:** Mebratu Abraha, Atkure Defar.

**Writing – original draft:** Mebratu Abraha.

**Writing – review & editing:** Mebratu Abraha, Getinet Ayano, Dereje Bayissa, Abraham Getachew, Mahteme Bekele, Melsew Getnet, Melaku Seyum, Atkure Defar, Sileshi Demelash, Gizachew Taddesse, Tariku Shimels.

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
