## [Decision Letter · Decision Letter 0]

7 Jul 2021

 PGPH-D-21-00058 Prevalence and correlates of depression and anxiety in quarantined population during the COVID-19 outbreak in central Ethiopia PLOS Global Public Health

Dear Dr. Abraha,

Thank you for submitting your manuscript to PLOS Global Public Health. After careful consideration, we feel that it has merit but does not fully meet PLOS Global Public Health’s publication criteria as it currently stands. Therefore, we invite you to submit a revised version of the manuscript that addresses the points raised during the review process.

 This is a relevant piece. The topic is important; quarantined persons have not received enough attention. The manuscript does require a major revision from the authors. Please consider both reviewers comments as they provide important questions and practical ways to address the manuscript's weaknesses.  

We look forward to receiving your revised manuscript.

Kind regards,

Cristian R. Montenegro

Academic Editor

Journal Requirements:

Additional Editor Comments (if provided):

1) Somewhere in the introduction, a timeline or reconstruction of key events in response to the pandemic in Ethiopia should be provided to help the reader better understand the context.

2) The article calls for attention to be given to mitigate mental health problems in the quarantined population during the COVID- 19 outbreak. One would expect a more concrete, evidence-based kind of suggestion, differentiated along the lines of the specific results and not a generical "more attention" type. Attention to what specifically? What kind of interventions could be developed?

3) Careful proofreading is required.

Reviewers' comments:

Reviewer's Responses to Questions

**Comments to the Author**

1. Does this manuscript meet PLOS Climate’s publication criteria? Is the manuscript technically sound, and do the data support the conclusions? The manuscript must describe methodologically and ethically rigorous research with conclusions that are appropriately drawn based on the data presented.

Reviewer #1: Yes

2. Has the statistical analysis been performed appropriately and rigorously?

Reviewer #1: Yes

3. Have the authors made all data underlying the findings in their manuscript fully available (please refer to the Data Availability Statement at the start of the manuscript PDF file)?

Reviewer #1: Yes

4. Is the manuscript presented in an intelligible fashion and written in standard English?

PLOS Climate does not copyedit accepted manuscripts, so the language in submitted articles must be clear, correct, and unambiguous. Any typographical or grammatical errors should be corrected at revision, so please note any specific errors here.

Reviewer #1: No

5. Review Comments to the Author

Reviewer #1: Many thanks for having the chance to review this article that aimed to explore the prevalence and factors associated with anxiety and depression among in quarantined population during the COVID-19 outbreak.

- Title should be shortened (including both prevalence and correlates of depression and anxiety) confuse the reader. Please be more specific.

Abstract

- Background part is very large. Please shorten.

- At the methods part few things are repeated. Please reformulate.

- At the conclusion part be more specific. Do not repeat the results section.

Introduction

- Please update information about Ethiopia

- Paragraph 1 and 2 should be merged.

- Paragraph 3 second sentence. You are saying about quarantine and after jumping directly to infectious out brakes. An infection outbreak doesn’t mean quarantine so please clarify this part.

- In general, the introduction part is very long. Similar things are repeated. Is better to shorten this part and to be more concise on what you want to say.

Methods

- Subsections at the methods part should be more concrete and no so general (i.e. Study population, sample size determination, sampling technique, and inclusion criteria). Please split these parts.

- Please give more information / clarify about “purposively selected the quarantine centers”.

- Its not very clear to me how researchers selected their sample;

- Did you conduct face validity for questions of socio-demographic characteristics?

- How were the researchers recruited?

- Who trained the researchers?

- Researchers were in pairs or individually during the process of data collection. Please give more information about this.

- It is not clear to me something. You used a structured questionnaire. Why you needed to have trained researchers for this and especially so many.

- What about rejection rates? Which were the reasons for this?

- Participants in quarantine. How many of them were infected? Did you have differences between them infected and non-infected?

- There is not the protocol number at the ethical approvement decision;

Results

- First sentence of the results section. The whole sentence is one paragraph. Please split it.

- What do this sentence mean (second sentence of results part). Please clarify in US dollars. “Also, 209 (70.4%) had 8000 Ethiopian birrs and below as a monthly income with a mean of 8783.74.

- Please correct the first sentence (put absolute numbers in parenthesis) of the section Clinical and psychosocial characteristics;

- As above on the next section. Please check this in the whole text;

- How are defined the stressful events;

- In general sentences are very long. Please split them because the reader loses the meaning;

- The four first sections of the results part could be merged. Two many information are provided that are also available in the tables;

- A table about bivariate analysis is needed;

- Is better to have in one table the adjusted and non-adjusted ratio rates;

- TO my view the focus should be on the multivariate analysis and not so much on the univariate. Too many information are given for the second but not the first.

Discussion

- Please clarify this “Sub-Saharan Africa including Ethiopia. Wasn’t the study conducted only in Ethiopia and more specifically in the capital.

- There are many other articles that says the opposite. “receiving different refreshing information and games through these social media’s as well as mass media’s could help the individuals to experience more relaxation and exposure with others. Please check this part.

- Last paragraph of the discussions is similar to the third paragraph. Please check.

- Which are the strengths and limitations of this study?

Conclusions

- Please mention implication for future research.

- English editing is needed for the whole text.

6. PLOS authors have the option to publish the peer review history of their article (what does this mean?). If published, this will include your full peer review and any attached files.

**Do you want your identity to be public for this peer review?** For information about this choice, including consent withdrawal, please see our Privacy Policy.

Reviewer #1: **Yes: **Enkeleint A. Mechili

**Comments to the Author**

1. Does this manuscript meet PLOS Global Public Health’s publication criteria? Is the manuscript technically sound, and do the data support the conclusions? The manuscript must describe methodologically and ethically rigorous research with conclusions that are appropriately drawn based on the data presented.

Reviewer #2: Yes

2. Has the statistical analysis been performed appropriately and rigorously?

Reviewer #2: No

3. Have the authors made all data underlying the findings in their manuscript fully available (please refer to the Data Availability Statement at the start of the manuscript PDF file)?

Reviewer #2: No

4. Is the manuscript presented in an intelligible fashion and written in standard English?

Reviewer #2: Yes

5. Review Comments to the Author

Reviewer #2: This is an important paper, being one of the first studies on the effects of quarantine on mental health in the Global South. The findings are partially expected, but leave some questions unanswered that need to be addressed. Also, methods and data analysis should be further fleshed out as noted below.

First, it is unclear how exactly quarantines work in Ethiopia. For example, what is the criteria to determine someone should quarantine? Testing positive for COVID-19? Having a close one who tested positive for COVID-19? Entering the country? All of them? That should be clearly stated so we know who the target population is.

Also, just wondering, did the government implement any quarantines affecting entire neighborhoods/communities? Were people from those communities also of interest to this study? Were those “total quarantines” or “partial quarantines” (e.g., only for the weekends)? You may want to operationalize what “being in quarantine” means here.

Second, the description of the target population is somewhat incomplete. Who were these people? How different were they compared to the general population? Were they at higher risk of reporting mental anxiety/depression? All this should be said explicitely (e.g., as hypotheses). The authors should note, though, that given we don’t know much about this population it’s challenging to say that they were at higher risk of mental symptoms. For instance, we don’t know if they had a prior mental/physical condition or if their parents had mental/physical conditions too, both of them strong predictors of mental issues.

Third, it seems the authors used the HADS plus a series of ad-hoc questions on, for example, social support, stressful life events, substance use history, etc. Please describe those questions/items at length. Furthermore, why did the authors collect data on those variables in particular? Were those hypothesized to be related to the exposure and the outcome (i.e., confounders)? If so, it should be mentioned in the Introduction/Background.

Fourth, the data analytic plan needs refinement. For example, how did the authors conceptualize the exposure/predictors, outcome, confounders, etc? It seems they put everything into a logistic model and saw what came up significant. This approach has been largely criticized in epidemiology because some of those variables could be considered indeed as confounders, but others could be effect modifiers and/or mediators (for which you don’t have to control for). Please provide a rationale for your analytic approach taking into consideration all this.

Fifth, the authors should distinguish between “crude” versus “adjusted” results, and say which variables they adjusted for (e.g., sociodemographic?). Also, please describe variables of interest before presenting the results. For example, the following paragraphs makes no sense if you haven’t defined each variable previously: “bivariate analysis was done and socio-demographic factors such as having a child, stressful life event, spending the quarantine time by sleeping, and perceiving transmission of COVID-19 can be prevented by wearing of glove for anxiety and factors such as marital status, stressful life event, method of refreshment, source of information about COVID- 19, perception on the cause of COVID-19, knowledge on method of prevention and utilization of COVID-19”

6. PLOS authors have the option to publish the peer review history of their article (what does this mean?). If published, this will include your full peer review and any attached files.

**Do you want your identity to be public for this peer review?** For information about this choice, including consent withdrawal, please see our Privacy Policy.

Reviewer #2: **Yes: **Franco Mascayano

---

## [Editor Report · Decision Letter 1]

2 Nov 2021

PGPH-D-21-00058R1

Depression and anxiety among quarantined population during the COVID-19 outbreak in central Ethiopia

Dear Dr. Abraha

**[Earlier today I have sent another email by mistake. Please ignore that message and consider this one instead]**

Thank you for submitting your manuscript to PLOS Global Public Health. After careful consideration, we feel that it has merit but does not fully meet PLOS Global Public Health’s publication criteria as it currently stands. Therefore, we invite you to submit a revised version of the manuscript that addresses the points raised during the review process.

We look forward to receiving your revised manuscript.

Kind regards,

Cristian R Montenegro

Academic Editor

Journal Requirements:

Additional Editor Comments (if provided):

Thanks again for the opportunity to help with this manuscript. Thanks to the authors for the hard work in making important improvements, and for the humility to take on board detailed and challenging reviewer comments and requests. Reviewers' points had been addressed in a convincing way, the manuscript is clearer and it can be of interest for a wider, international audience. More contextual elements are introduced and the methodology has been further specified.

The piece still needs improvements in terms of English grammar before it's ready for publication.
---

## [Editor Report · Decision Letter 2]

18 Nov 2021

Depression and anxiety among quarantined population during the COVID-19 outbreak in central Ethiopia

PGPH-D-21-00058R2

Dear Dr. Abraha,

We're pleased to inform you that your manuscript has been judged scientifically suitable for publication and will be formally accepted for publication once it meets all outstanding technical requirements.

Within one week, you'll receive an e-mail detailing the required amendments. When these have been addressed, you'll receive a formal acceptance letter and your manuscript will be scheduled for publication.

An invoice for payment will follow shortly after the formal acceptance. To ensure an efficient process, please log into Editorial Manager at https://www.editorialmanager.com/pgph/ click the 'Update My Information' link at the top of the page, and double check that your user information is up-to-date. If you have any billing related questions, please contact our Author Billing department directly at authorbilling@plos.org.

Kind regards,

Cristian R Montenegro

Academic Editor

Additional Editor Comments (optional):

Thanks for taking onboard the comments and suggestions and for persevering with the manuscript. 